# Social Inequity and Wildfire Response: Identifying Gaps and Interventions in Ventura County, California

Brianna Baker, Yvonne Dinh, Iris R. Foxfoot, Elena Ortiz, Alison Sells and Sarah E. Anderson *

Bren School of Environmental Science & Management, University of California Santa Barbara, Santa Barbara, CA 93106, USA; brianna_baker@bren.ucsb.edu (B.B.); foxfoot@bren.ucsb.edu (I.R.F.)
* Correspondence: sanderson@bren.ucsb.edu

**Abstract:** As climate change increases the frequency and severity of wildfires across the Western U.S., there is an urgent need for improved wildfire preparedness and responses. Socially marginalized communities are particularly vulnerable to wildfire effects because they disproportionately lack access to the resources necessary to prepare for and recover from wildfire and are frequently underrepresented in the wildfire planning process. As an exemplar of how to understand and improve preparedness in such communities, this research identified communities in Ventura County facing heightened marginalization and risk of wildfire using spatial analysis. Researchers then deployed a county-wide survey and held focus groups in two communities identified in the spatial analysis. Research revealed that non-English speakers, women, people of color, and newer residents in Ventura County are less prepared for wildfire than other groups. Based on these findings, this paper recommends an expansion of traditional risk mitigation programs, strengthened community engagement efforts, and strategies that increase community resources and leadership to decouple marginalization and wildfire vulnerability.

**Keywords:** diversity; equity; environmental justice; inclusion; future-thinking; well-being; wildfire; workforce; labor/labour; social vulnerability; planning

## 1. Introduction

Communities acutely need equitable wildfire preparedness and response. Climate change-induced extreme fire weather is common across the Western U.S., where wildfires burn increasingly large areas [1,2]. Marginalized communities are particularly vulnerable [3–8] because systemic inequities including poverty, poor vehicle access, and crowded households lead to disparities in their ability to respond [9]. While it is known that social, geographic, and biophysical factors compound the likelihood of exposure to the negative outcomes of wildfire [10,11], the particular needs of marginalized communities are difficult for external researchers and public officials to identify. This manuscript demonstrates the use of mixed methods—a survey and focus groups—in partnership with established community groups to identify the specific needs of marginalized communities. Then, it identifies strategies for addressing those needs. To ensure community resilience, wildfire preparation and planning must attend to social inequities and meaningfully support communities that have been marginalized [4–6].

Social inequities brought by marginalization are a driving force behind social vulnerability [12]. Social vulnerability to environmental hazards is characterized by a lack of access to resources, including political power or representation, infrastructure and social support, and income and wealth [10]. Social, economic, and political structures lead to marginalization by causing harm and limiting resource access among certain groups of people. These established systems create and perpetuate "inequity", or the unjust disparities in social and economic outcomes along racial, economic, age, and gender divides, among others. The harm experienced can be compounded by overlapping identities [4,5,10].

People with identities marginalized by these systems often lack necessary resources and, therefore, experience social vulnerability to wildfire. In summary, unjust social structures that exclude certain groups (marginalization) lead to disparities in social and economic outcomes for these groups (inequity), which can make it harder for them to prepare for, respond to, or recover from environmental hazards such as fire (vulnerability).

The influence of marginalization on individuals' susceptibility to harm disrupts the conception of disasters as "natural" events, since disproportionate social impacts result from established systems created by people and society [6,13]. In wildfire, the influence of marginalization on vulnerability plays out before, during, and after a community experiences a wildfire. Income, poverty, race, language, education, disability, age, vehicle access, and caregiver support, among others, shape individual and community vulnerability to wildfire by affecting their ability to prepare, evacuate, and recover [4,5,8,14]. The experience of each marginalized identity is different and can influence an individual's vulnerability across the temporal experience of wildfire, such that they experience multiple effects [8,10]. While the underlying marginalization that makes up social vulnerability may not be within the control of wildfire managers, vulnerability to wildfire can be mitigated.

Wildfire managers must decouple vulnerability to disaster from social marginalization by addressing social vulnerability in wildfire planning and purposefully including marginalized communities. Approaches include the direct provision of resources to vulnerable groups to overcome inequities. An example is Central Coast Alliance United for a Sustainable Economy's (CAUSE) 805 UndocuFund, which supported undocumented migrant workers and their families impacted by the Thomas Fire [15]. Wildfire managers can also disrupt patterns of social vulnerability by ensuring planning efforts and resource allocation include or are led by those disproportionately impacted by fire through community advisory bodies or more inclusive hiring practices and work culture. We further explore examples and possible solutions in Section 4.2.3. While societal-level change to end the root causes of inequity and vulnerability must continue beyond the wildfire space, intermediary efforts to address vulnerability in the field are necessary.

Ventura County, California exemplifies a fire-prone landscape where social marginalization and vulnerability are linked and increased fire frequency and severity due to climate change threatens county residents (Figure 1) [16]. As such, it serves as a useful case study. In Ventura County, 6.1% of families live in poverty and 4.3% of households do not have access to a car. Approximately 15% of the population is over 65, and 10.9% of the population has a disability. Over nine percent face language barriers, exceeding the national average [17]. At the end of the 2022 fire season, two of the 20 most destructive fires in California's modern history were in Ventura County: the 2017 Thomas Fire and the 2018 Woolsey Fire [18]. During the Thomas Fire, language barriers stymied the distribution of emergency response information and predominantly Latine and Indigenous farmworkers were exposed to unhealthy levels of smoke at work. After the fire, people who commute out of the county for employment experienced disruptions to transportation and housing. Many residents were barred from receiving government disaster aid due to their citizenship status [5]. The effects of the Thomas Fire emphasize the need to attend to the disproportionate impacts of wildfire on marginalized communities [19–21].

This research, which stems from a partnership between Ventura Regional Fire Safe Council and student researchers at the University of California, Santa Barbara, aims to better understand the disproportionate impacts of wildfire in Ventura County and outline possible solutions. The project was initiated by Ventura Regional Fire Safe Council, which sought recommendations to address social vulnerability in their Community Wildfire Protection Plan update. While project conceptualization, execution, and analysis of data were primarily completed by the researchers, Ventura Regional Fire Safe Council attended several scoping meetings, reviewed drafts of the survey and the focus group agendas, and connected the researchers to other community organizations.

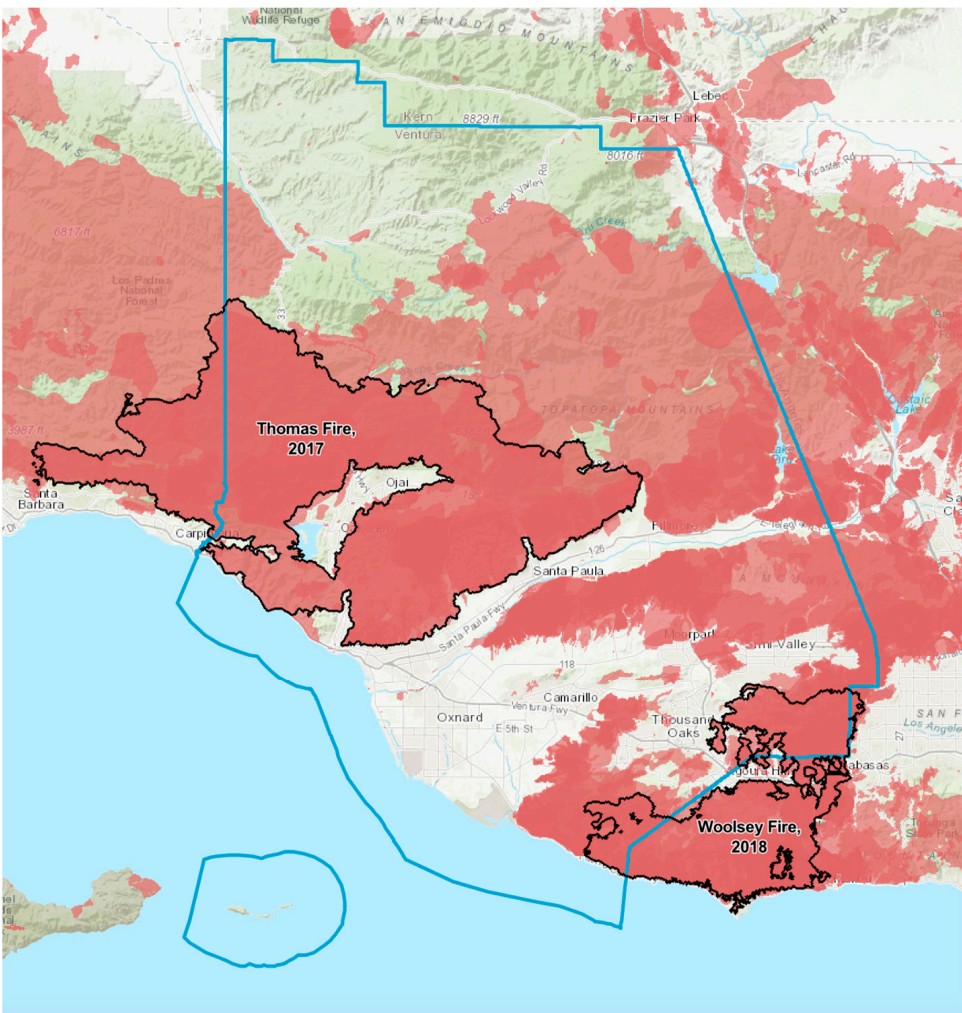

**Figure 1.** CalFire perimeter data (in red) of fires in Ventura County (blue line) since 1950.

We implemented a mixed methods approach to (1) identify vulnerable populations, (2) synthesize community feedback, and (3) produce policy and community engagement recommendations. Existing spatial data were used to identify where socially vulnerable groups coincide with biophysical fire threat in Ventura County. A survey of Ventura County residents illustrated resident perspectives on wildfire, including concerns and barriers to preparation (Supplementary Information Section A. The survey responses showed that socially vulnerable groups face additional challenges relating to wildfire preparation and response. Two focus groups in the central part of the county illuminated the specific lived experiences of people at the intersection of socially vulnerable communities and high biophysical threat of wildfire. The collective findings reveal that needs and barriers related to wildfire safety for Ventura County residents differ based on social identity and residence time.

In addition to supporting Ventura County-specific wildfire risk management, the methods of engagement and lessons learned from this project act as a model for other communities living with wildfire. Communities in California and beyond can refer to these methods for identifying vulnerable populations and addressing social marginalization concerns in their planning processes. This work contributes to continued efforts to make community wildfire planning, and disaster planning more broadly, responsive to the vulnerabilities of socially marginalized communities.

## 2. Materials and Methods

### 2.1. Spatial Analysis

We conducted a spatial analysis of wildfire risk and social vulnerability factors to identify socially vulnerable census tracts at high risk of wildfire in Ventura County. Wildfire risk is the "likelihood, intensity, and susceptibility to effects of wildfires on highly valued resources and assets" [22]. This is based on biophysical characteristics, such as fuel, weather, and topography, as well as an evaluation of infrastructure and community assets at risk. A raster layer produced by the U.S. Forest Service called Risk to Potential Structures (RPS) represented biophysical risk. Each cell represents the likelihood of wildfire and intensity of wildfire-related risk to a structure at a given location [23].

The U.S. Centers for Disease Control and Prevention's (CDC) social vulnerability index (SVI) was used to map social vulnerability. This layer considers fifteen variables as indicators of socioeconomic vulnerability to disasters. The aggregate SVI score that considers all measures is the basis of the analysis [24]. Individual census tracts are ranked relative to others by indicator variable, and the rank for each variable is used to create a total vulnerability score. The SVI is displayed by census tract, along with the average RPS per census tract found by using the Spatial Analyst toolbox in ArcGIS Pro (Figure 2).

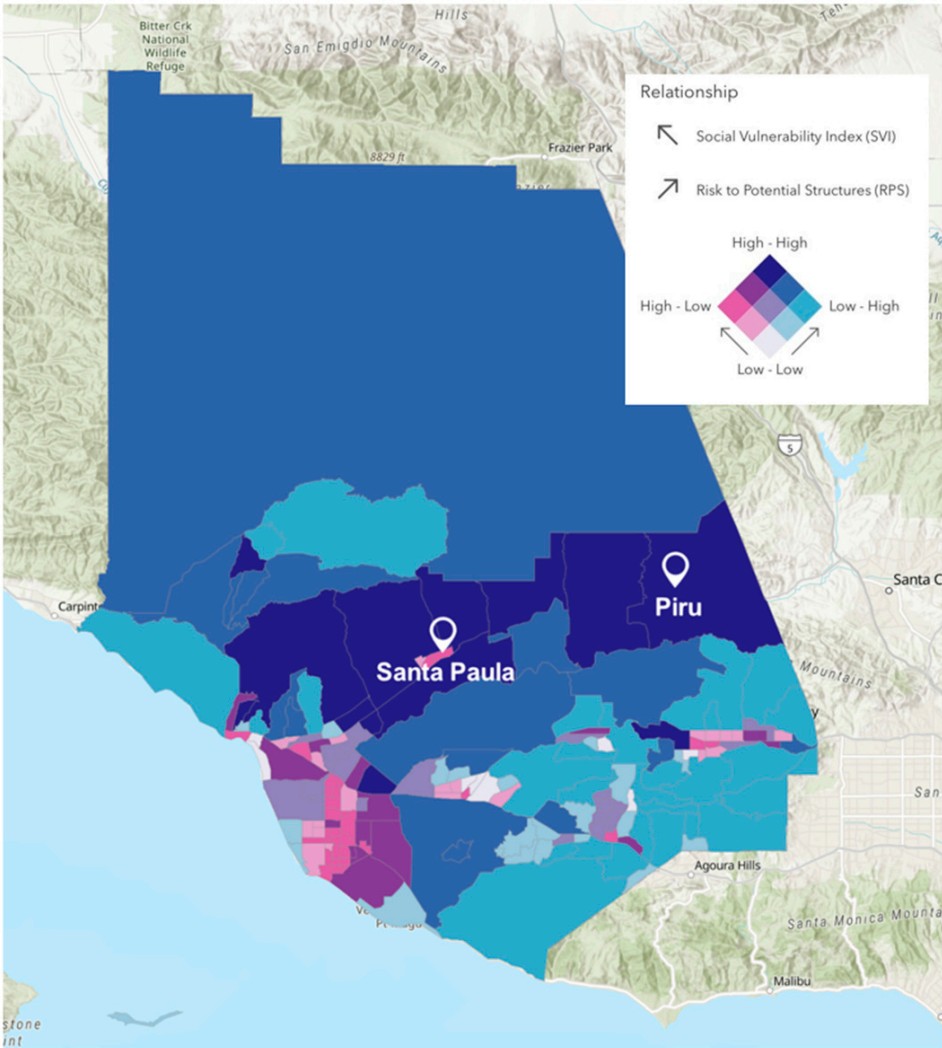

**Figure 2.** Map of focus group locations, Santa Paula and Piru, overlaid on Social Vulnerability Index (SVI) and Risk to Potential Structures (RPS) layers. Dark indigo indicates areas with high SVI scores and high RPS.

This analysis identified areas that were both high in wildfire risk and high in social vulnerability, particularly areas along the central portion of the county. The spatial intersection of wildfire risk and social vulnerability informed the placement of focus groups.

### 2.2. Survey and Survey Analysis

A survey of Ventura County residents' perceptions of and concerns related to wildfire quantitatively investigated how race, class, gender, age, language, and mobility can compound and impact people's ability to prepare for, respond to, cope with, and recover from wildfire. Although our community partners at Ventura Regional Fire Safe Council reviewed survey questions and we tested the survey on a convenience sample, no pilot testing of the survey with the target population was conducted. To address potential barriers to participation for those with higher social vulnerability factors, the research team distributed the survey with the help of established community organizations in socially vulnerable areas of the county. Ventura Regional Fire Safe Council (Ventura Fire Safe) and other local community organizations, including Ventura County Fire Department, Mixteco Indigena Community Organizing Project (MICOP), Promotoras Y Promotores Foundation, Piru Community Council, Westside Community Council, and other local Fire Safe Councils circulated the survey online. Available 27 August 2021 to 9 January 2022 via Qualtrics in both English and Spanish, the electronic survey was disseminated via email, Ventura Fire Safe's website, and social media pages (such as Instagram). Ventura Regional Fire Safe Council collected a small number of paper surveys at a community event, and community partners distributed paper surveys to predominantly Spanish-speaking residents. As an incentive to participate, respondents could enter a raffle for one of five USD25 Visa gift cards.

The survey included six questions related to social demographics, seven questions on wildfire experience and response, and six questions related to the Community Wildfire Protection Plan (CWPP) and wildfire planning. In total, 489 people responded. We eliminated responses in which the respondent declined to answer all questions of interest (67) or did not have a Ventura County zip code (18), leaving 404 survey responses for analysis.

To assess whether the survey demographics were representative of Ventura County, chi-squared tests of age, income, race, gender, and language spoken at home were used to compare the sample against the demographics reported in the American Community Survey (2019). Chi-squared tests confirmed that the survey sample was not representative of Ventura County, yielding *p*-values of less than 0.05 for all variables (age, income, race, gender, and language spoken at home).

Among the survey respondents, 90% speak English at home and 78% identify their race as white alone. Most respondents identify as women (73%) and 39% of respondents are over the age of 65. According to the 2019 American Community Survey, women make up 51% of the population, and 21% of Ventura County is over 65 (Table 1).

A raking method was employed to weight the survey results proportionally to the population of Ventura County. Following the American National Election Study methodology and the accompanying R package "anesrake" [25], each survey response was assigned a weighting factor based on the particular demographics of the respondent. These weighted data served as the basis for all subsequent analyses.

Many respondents chose not to disclose their income, which resulted in a large amount of missing data, likely in a non-random pattern. We imputed the missing data using the predictive mean matching method in the mice package in R 4.0.3 [26]. This method predicts the value of the missing variable using regression, then randomly selects a replacement value from five observations that are most similar to the predicted missing value. To decrease random variation, the process is then iterated 25 times and the results are pooled. We present three variations for each model: one without income, one with imputed income, and one with non-imputed income, which is missing 119 entries.

**Table 1.** Comparison of Ventura County demographics with survey respondent demographics.

| Demographic Variable | Proportion in ACS 2019 | Proportion in Survey | *p*-Value (From Chi-Squared Test) |
|---|---|---|---|
| Gender | | | |
| Woman | 0.51 | 0.73 | $p < 0.05$ |
| Man | 0.49 | 0.27 | |
| Age | | | |
| 18–24 | 0.12 | 0.07 | $p < 0.001$ |
| 25–34 | 0.17 | 0.07 | |
| 35–44 | 0.16 | 0.09 | |
| 45–54 | 0.17 | 0.18 | |
| 55–64 | 0.17 | 0.27 | |
| 65–74 | 0.12 | 0.27 | |
| 75+ | 0.09 | 0.12 | |
| Race | | | |
| American Indian or Alaska Native | 0.01 | 0.01 | $p < 0.001$ |
| Asian | 0.08 | 0.03 | |
| Black or African American | 0.02 | 0.01 | |
| Native Hawaiian or Pacific Islander | 0.001 | 0.002 | |
| Two or more races | 0.05 | 0.16 | |
| White | 0.84 | 0.78 | |
| Household annual income | | | |
| Less than USD10,000 | 0.03 | 0.05 | $p < 0.001$ |
| USD10,000–14,999 | 0.02 | 0.02 | |
| USD15,000–24,999 | 0.05 | 0.01 | |
| USD25,000–34,999 | 0.05 | 0.02 | |
| USD35,000–49,999 | 0.09 | 0.06 | |
| USD50,000–74,999 | 0.14 | 0.13 | |
| USD100,000–149,999 | 0.15 | 0.24 | |
| USD150,000–199,999 | 0.20 | 0.15 | |
| USD200,000 or more | 0.12 | 0.19 | |
| Language spoken at home | | | |
| English | 0.61 | 0.90 | $p < 0.001$ |
| Not English | 0.39 | 0.10 | |

Three survey questions served as the basis for additive indices. The first index, hereafter referred to as the "wildfire impacts index", represented the number of ways that the respondent had experienced wildfire and was compiled by asking "In what way(s) has your household been affected by wildfire? Check all that apply". The second index, the "worries index", represented the number of concerns the respondent expressed about wildfire. It was measured by asking "If you worry about wildfire, what concerns you most? Check all that apply". The final index, "barriers to evacuation index", represented the number of barriers to evacuation that the respondent reported facing in response to the question: "Is there anything that would make it difficult for your household to evacuate during a wildfire? Check all that apply." (Supplementary Information Section A). Each checked response was given a value of one, except the null responses ("I have not been affected by wildfire", "I do not worry about wildfire", and "No, I could easily evacuate"), which were given a value of zero. In all cases, the response of "Other" was scored a value of one, even when the respondent selected "Other" and wrote in more than one additional answer in the provided text box. These values were summed to produce an additive index value for each of the three questions. Principal component analysis revealed that variance in all three indices is irreducible, indicating that there is no mutual underlying factor between potential responses. As such, we conclude that each potential response should be considered as an individual factor, justifying the use of a simple additive index.

We performed ordinal logistic regressions to assess how evacuation preparedness, wildfire concerns, and evacuation barriers differ among demographic groups in Ventura County [27]. The three dependent variables were responses to the question "Do you currently feel prepared to evacuate your home in the event of a wildfire? (responses: no, somewhat, and yes) and two of the indices outlined above. Home insurance status and pet

ownership did not contribute to model fit. Age, gender, mobility concerns, language, years living in Ventura, race, and income provided the best overall model fit. The U.S. Forest Service's Mean Risk to Potential Structure (RPS) [23] values per zip code served as a control for wildfire hazard and the wildfire impacts index controlled for prior experience with wildfire and evacuation.

### 2.3. Focus Groups

The research team also hosted focus groups with people living in census tracts with high social vulnerability that were at high risk of wildfire, to better elucidate the lived experiences of these communities that may be missed in census or survey data. Community partners in the region helped host two focus groups in central Ventura County (Figure 2): an English conversation in Piru and a Spanish conversation in Santa Paula. In Piru, five residents participated for 45 min. In Santa Paula, 12 residents participated for one and a half hours. Community organizations who partner with Ventura Fire Safe connected the research team with residents; therefore, the conversations took place in established communities of people where some trust and relationships already exist. These organizations' members have low incomes, are predominantly Spanish speakers, or have some experience with community organizing efforts in the Central Coast region.

The conversations centered around three main questions to allow residents to steer the dialogue based on needs and interests: (1) what is working well with wildfire prevention and response?; (2) what is missing?; and (3) how can Ventura Fire Safe support communities and fill in gaps? A full list of questions is located in Supplementary Information Section B. These themes were discussed with Ventura Fire Safe, but there were no efforts to conduct pilot focus groups ahead of time. Instead, the focus groups documented below are meant to be pilot studies for future collaborative efforts.

The team obtained informed consent from participants through a consent form that transparently communicated goals, data use, and the risks of participation, and through an oral explanation with the opportunity to ask questions in-person prior to the session. All participants were compensated for their time with USD25 Visa gift cards. The notes from the focus groups were analyzed by the research team to identify major themes and ideas.

## 3. Results

The results of this mixed methods approach revealed the differing needs, concerns, and lived experiences of Ventura County residents in relation to wildfire events. The survey broadly elucidated how residents perceive, prepare for, and live with wildfire. The focus groups revealed the sentiments of the target demographic groups—residents from low income, Spanish-speaking households, and living in high wildfire risk areas. At times, the survey results conflicted with data collected from the focus groups, which underscores how different tools reach different populations and that the target population's needs differ from the broader community.

### 3.1. Survey Analysis

The analysis of the survey data centered on the themes of evacuation preparedness and wildfire risk mitigation. This is because the project was undertaken in service to the Ventura Regional Fire Safe Council, who were particularly concerned with understanding whether residents are prepared and what barriers exist. Many (55%) of the survey respondents had evacuated from wildfires in the past (Figure 3). Six people reported that they wanted to evacuate but could not. Commonly reported effects of wildfire, aside from evacuation, included impacts on well-being and stress and impacts from smoke. Results suggest that wildfire is a salient issue for county residents and imply that the impacts go beyond the threat of the flame front itself.

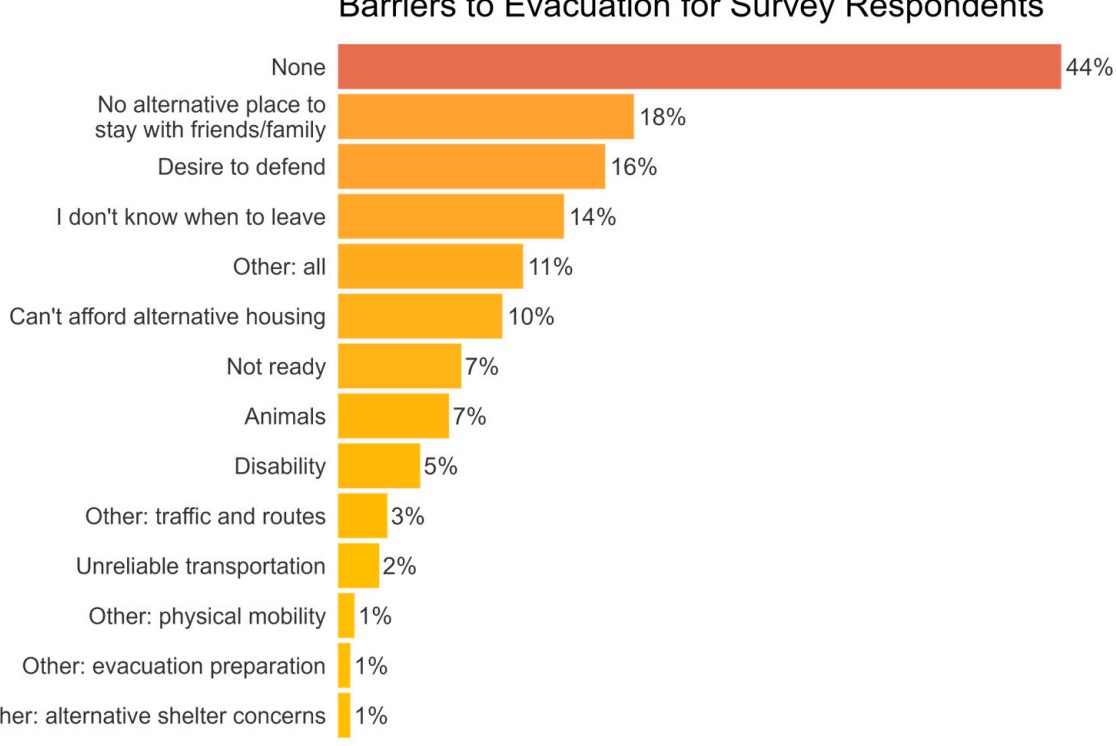

**Figure 3.** Past effects of wildfires on Ventura County residents surveyed. n = 404 respondents.

Barriers to evacuation were wide-ranging. Forty-four percent of survey respondents reported that they could easily evacuate (Figure 4), while other residents reported impediments. Commonly cited barriers to evacuation included the lack of an alternative shelter, the desire to stay and defend property, and a lack of information about when evacuation was necessary. This broad range of experiences could be a result of the range of survey respondents' social demographics.

**Figure 4.** Barriers to evacuation for surveyed Ventura County residents. Barriers noted as "other" were not included in the survey but were written in by individual survey respondents and categorized. n = 404 respondents.

Many respondents wrote in additional barriers to evacuation beyond options provided in the survey. Twelve respondents were concerned about traffic and a lack of alternative evacuation routes and four reported physical mobility challenges. At the household level, one respondent noted the loss of food and the costs associated with evacuation, which is a particular issue for families experiencing food insecurity. Another worried about the impacts on their child with autism. These results show how a wildfire event can impact various facets of daily life.

The ordinal logistic regression for evacuation preparedness that included the variables of age, gender, mobility concerns, language, time spent in Ventura, race, previous experience with evacuation, and income provided the best overall model fit for the survey data (Supplementary Information Section C, Tables S1–S3). In general, the three models (no income, income imputed, income included but fewer observations) indicated similar results, except for the independent variables of gender and language spoken at home. Income was not associated with evacuation preparedness in the models including stated income and imputed income (Figure 5).

**Figure 5.** Coefficient plot of three best models for evacuation preparedness. Note that, for categorical variables, each coefficient estimate is relative to a reference category. The reference category for age is 18–24, the reference category for gender is "Man", the reference category for mobility issues is "no mobility issues", the reference category for language is "English alone", the reference category for years in Ventura is "one year or less", and the reference category for race is "white alone". RPS stands for Risk to Potential Structures and is a measure of the physical risk of a wildfire. Certain groups like long-time Ventura residents, men, and bilingual people were more likely to feel prepared to evacuate. In contrast, women, people with mobility issues, and people of color were less likely to feel prepared to evacuate.

There were significant differences in stated evacuation preparedness across age groups, gender, and income levels (Figure 5). Residents aged 18–24 years old were least likely to indicate they are prepared to evacuate compared to other age groups, although 45–54 year-olds were not significantly different. Respondents identifying as women were less likely to indicate evacuation preparation than men. Those who indicated their gender identity as non-binary, other, or who prefer not to disclose their gender were even less likely than women to indicate preparation among those willing to provide income information, in which there was no relationship. Across all models, respondents identifying as a person of color (race was anything other than white alone) were less likely to indicate they were prepared than respondents identifying as white.

Non-English speaking and bilingual survey respondents generally reported higher evacuation preparedness than English speakers. But, because there were so few responses from non-English speakers (nine), these models may not accurately reflect this population. In all models, people who indicated they spoke English and another language were more likely to report being prepared than those who spoke English alone. In two models, non-English speakers were more likely to report being prepared than those who spoke English alone. However, among respondents who were willing to provide income information, non-English speakers were less likely to be prepared to evacuate. In the survey, two out of nine non-English speakers indicated they were not prepared to evacuate, four indicated they were somewhat prepared, and three indicated they were prepared to evacuate.

The results indicated the most dramatic differences in evacuation preparedness based on residence time in Ventura. Consistently across all models, people who had lived in the county for more than 10 years had between 3.35 and 4.23 higher log odds of reporting evacuation preparedness than people who had lived there for less than one year. Long-term residents reported being more prepared to evacuate, even controlling for previous evacuation experience. This likely reflects familiarity with evacuation routes, community resources, and established social networks, rather than direct past experience.

As Figure 6 shows, evacuation barriers varied across demographics, with lack of an adequate alternative shelter emerging as the most common hurdle. Many people with mobility issues indicated that their disability is a barrier. Many non-English speakers and recently established Ventura County residents noted a lack of information about when to evacuate. Not having an alternative place to stay in the event of an evacuation was also an issue for many non-English speakers. Men were more likely to state the desire to stay and defend their home, stalling evacuation. Lack of transportation, both for respondents or their animals, was not a major barrier to evacuation.

The ordinal logistic regressions of the evacuation barriers index (Figure 7) indicated that people with mobility issues were significantly more likely to report facing multiple barriers to evacuation, while bilingual individuals and long-term residents were more likely to report facing fewer barriers. These models reinforce that systematic challenges to evacuation are associated with certain demographic groups. Living in a zip code associated with higher mean risk to potential structure values was associated with a decrease in the number of barriers reported (Supplementary Information Section C, Tables S4–S6).

Although certain groups report facing additional barriers to evacuation, concerns about wildfire afflicted Ventura County residents more evenly. Ordinal logistic regressions with the wildfire concerns index as the dependent variable indicate that women and people with higher past wildfire impacts index scores had significantly more concerns (Figure 8; Supplementary Information Section C, Tables S7–S9).

Survey respondents were particularly supportive of proactive fire mitigation practices undertaken by the government and utility companies, such as electrical infrastructure maintenance (61% of respondents) and arboreal work (60%). Other highly desired actions included evacuation preparation (59%) and community emergency planning (60%). One respondent suggested developing trainings in renter communities to prepare them to help each other evacuate and learn safety strategies if evacuation is not an option. Some respondents mentioned considering controlled burning informed by Indigenous practices.

These responses demonstrate the depth of residents' understanding of wildfire risk and possible community interventions.

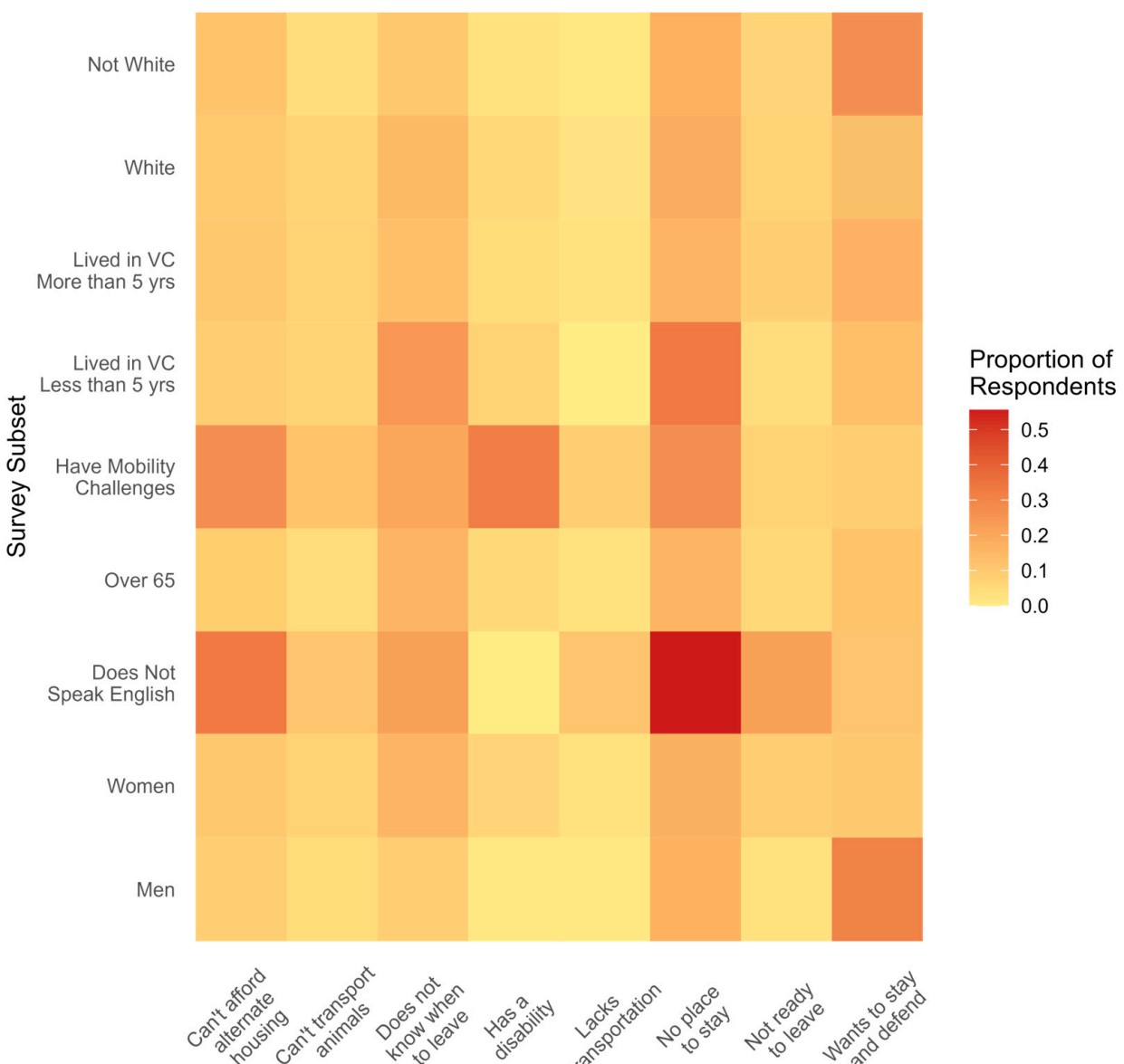

**Figure 6.** Barriers to evacuation for demographic subgroups. n = 404 respondents.

Wildfire mitigation preferences vary across Ventura County demographic subsets (Figure 9). Non-English speakers commonly selected actions at the household scale, such as home hardening and gutter cleaning, as top priorities for wildfire safety actions. Other demographic groups prioritized community emergency planning and electrical infrastructure maintenance, which are broad-scale actions unlikely to require individual action. This could be due to differing perceptions of the efficacy of mitigation strategies or could relate to a historical reliance on individual-level risk mitigation activities among non-English speakers, due to exclusion from broader-scale activities.

Social marginalization and vulnerability to wildfire are linked, and the intersections of social identity and vulnerability feed into limitations of the models. The models that include non-imputed income exclude respondents who chose not to disclose their income. There is likely a systematic reason that people chose not to report their income; those at the extremes

of the income scale may be uncomfortable disclosing their income. Non-English-speaking respondents who did not report their income also reported being prepared to evacuate. Omitting these responses results in a bias that understates evacuation preparedness among non-English speakers based on survey responses.

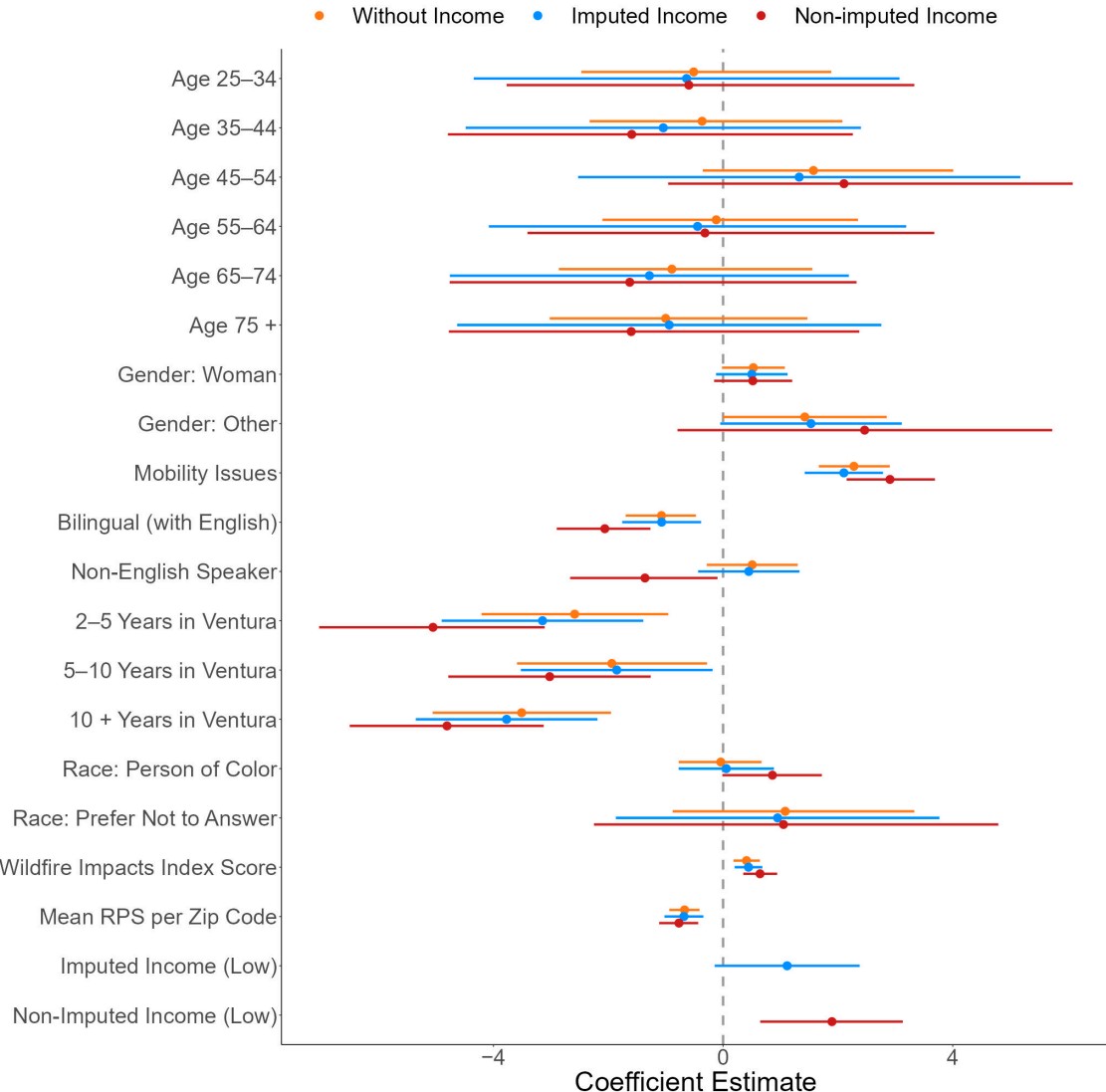

**Figure 7.** Coefficient plot of three models for barriers to evacuation index. Note that, for categorical variables, each coefficient estimate is relative to a reference category. The reference category for age is 18–24, the reference category for gender is "Man", the reference category for mobility issues is "no mobility issues", the reference category for language is "English alone", the reference category for years in Ventura is "one year or less", and the reference category for race is "white alone". RPS stands for Risk to Potential Structures and is a measure of the physical risk of a wildfire. Certain groups like non-mobility challenged people, long-time Ventura residents, bilingual people, and people living in high risk areas reported facing fewer barriers to evacuation. In contrast, people with mobility issues, recent Ventura County transplants, and people living in lower risk areas of Ventura County were more likely to report facing more barriers to evacuation. Age, gender, race, and non-English speaking were found to be insignificant in these models.

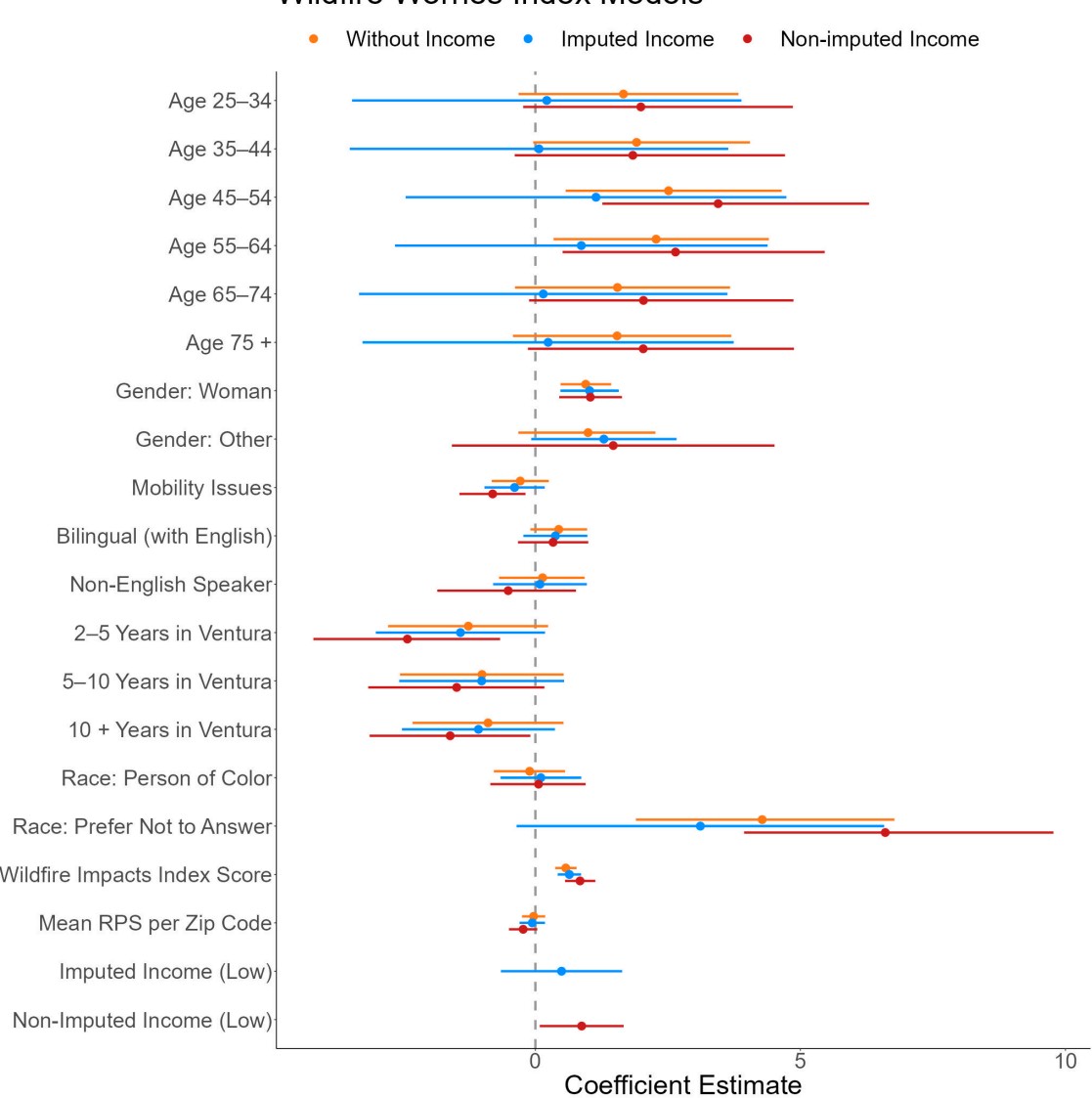

**Figure 8.** Coefficient plot of three models for wildfire-related worries index. Note that, for categorical variables, each coefficient estimate is relative to a reference category. The reference category for age is 18–24, the reference category for gender is "Man", the reference category for mobility issues is "no mobility issues", the reference category for language is "English alone", the reference category for years in Ventura is "one year or less", and the reference category for race is "white alone". RPS stands for Risk to Potential Structures and is a measure of the physical risk of a wildfire. Women were more likely to report having more concerns regarding wildfire than men. People with higher past wildfire impacts index scores (indicating people who have experienced more effects of wildfire in the past) were also more likely to report more worries regarding wildfire. Most other demographic factors, i.e., age, language, race, wildfire hazard exposure, and income, were found to have no significant association with the number of worries faced.

The research team encountered additional limitations in the survey data for non-English speakers. The small sample size (nine) of non-English speaking survey participants made it harder to conduct a robust statistical analysis or draw inferences. The results also demonstrate the limitations of using a written survey to reach these communities. The language question only asked participants to state the language(s) they spoke at home, not their preferred language. This may have incorporated speakers who are most comfortable communicating in a language other than English into the bilingual category, muddling the

relationships between language and wildfire risk. Alternative methods, such as the focus groups, appear to be more effective methods for targeting and meaningfully engaging non-English speakers.

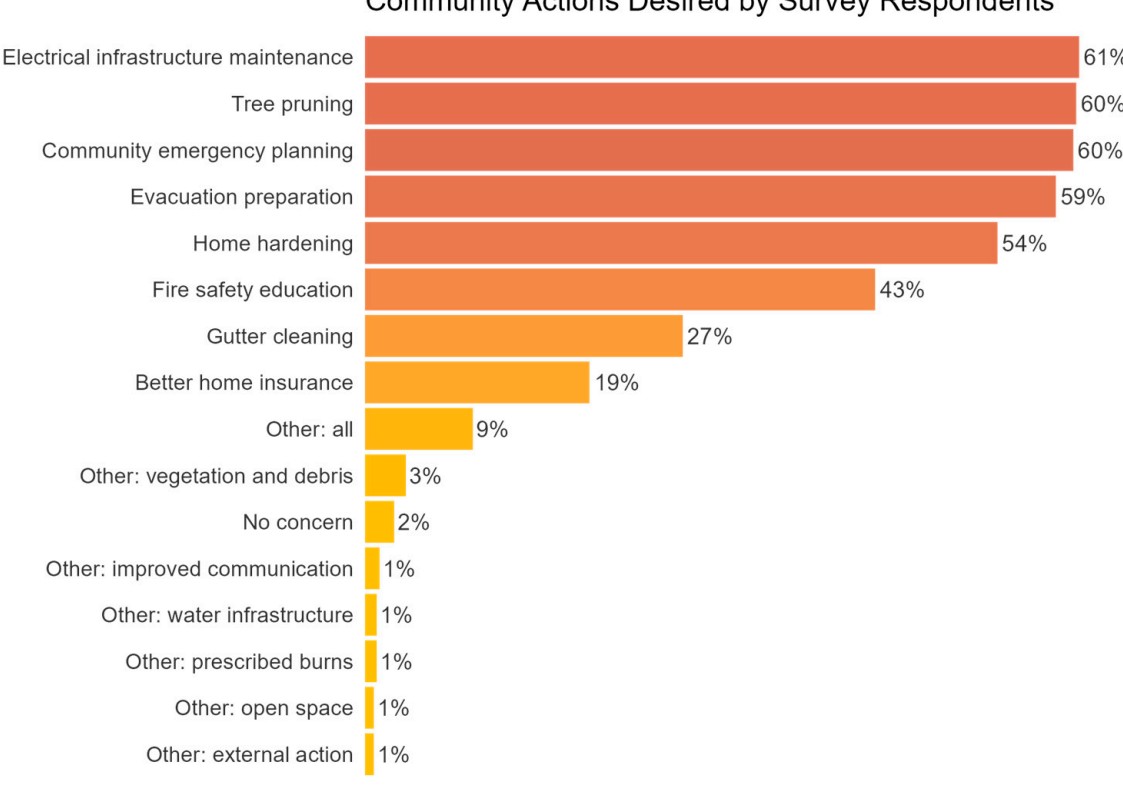

**Figure 9.** Desired community actions to reduce and mitigate wildfire. Community actions noted as "other" were not included in the survey but were written by individual survey respondents. n = 404 respondents.

### 3.2. Focus Groups

Focus groups centered on target demographic groups—low income, Spanish speaking, and living in high wildfire risk areas—who were not as well-represented in the survey. Both focus group conversations revolved around two themes: strengths and gaps in current wildfire prevention and response, and how Ventura Fire Safe could support communities and fill in gaps. During the focus groups, residents described their lived experience with wildfire, discussed barriers to resilience, and ideated possible points of intervention.

Focus groups engaged participants from distinct social demographic groups. In Piru, participants were mostly older men who were homeowners and from a majority lower to middle income community. Most residents had long-standing ties to the community and had lived there for many years. The Santa Paula focus group engaged mostly middle-aged women (72%) with children. Several of the participants mentioned they are involved with Indigenous and Latine organizing efforts in Ventura County. Others participate in "promotora" or community health worker organizing efforts locally. All participants in Santa Paula were native Spanish speakers.

Participants in both focus groups were particularly concerned about evacuation. In Santa Paula, residents expressed that many families do not have the vehicle capacity to evacuate all their children at once, necessitating multiple trips. One resident mentioned they were concerned about losing their job should they evacuate to Oxnard and not return to work the next day, making the decision to evacuate their family all the more difficult. Additionally, the entire valley has limited evacuation routes, challenging residents' ability

to get out quickly. This was a particular concern in Piru, where there are only two roads in or out. Both groups expressed the need for evacuation and shelter-in-place planning.

The focus groups also called for improved communication strategies, including requests for physical signs and flyers in Piru, and more Spanish and Mixteco communications in Santa Paula. Piru participants noted that older residents may prefer non-digital communication about wildfire education and risk mitigation. Santa Paula participants noted insufficient Spanish resources on wildfire prevention, evacuation, and response, including emergency notifications. They also noted that some older adult and Indigenous members of their community have difficulty accessing information because wildfire notifications are primarily dispersed via cell phones and social media, or text-heavy written materials. They reported that many Indigenous community members do not speak English or Spanish fluently and emphasized the need for more visual resources. Despite these limitations, participants in both groups noted that they use personal social networks to notify friends, family, and neighbors about acute wildfire threats using phone applications, such as WhatsApp, and speaking with neighbors in person.

The Santa Paula focus group expressed explicit concern for the effect of wildfire on mental health, particularly the mental and emotional well-being of children in their community. They shared that their children exhibit fear and anxiety in response to nearby wildfire events that do not directly impact their community. They felt that the psychological resources at school were insufficient in addressing their needs. One woman acknowledged that she, too, felt traumatized by past wildfire events. These anxieties relate to both immediate wildfire response and to residents' ability to successfully recover. In contrast, Piru residents focused on biophysical risks rather than the mental or emotional burden of fires.

As mentioned in the Santa Paula focus group, poor air quality is a particular problem for farm workers who are forced to choose between working in the smoke to keep their jobs and feed their families or losing income. The Thomas and Woolsey Fires both destroyed the homes of many Ventura County residents and brought hazardous air quality for extended periods. While both fires occurred several years ago, focus group participants remembered these wildfire effects keenly.

## 4. Discussion

### 4.1. Research Findings

Needs and barriers related to wildfire safety for Ventura County residents are influenced by social identity and residence time, which has implications for wildfire management and planning efforts. The focus groups revealed that low to middle income and non-English speaking people in Ventura County have differing needs, perceptions, and concerns regarding wildfire compared to the broader population. Specifically, they reported barriers to receiving important wildfire information, safely evacuating, and recovering from wildfire events compared to most county residents represented through the survey results. Additionally, they raised concerns around language access and mental health. Current wildfire planning and management do not equitably prepare all residents for wildfires, as they fail to address the needs of non-English speakers, women, communities of color, and newer residents who are shown to be especially vulnerable to wildfire impacts. In fact, interventions that are broadly applicable may systematically exclude residents who are particularly vulnerable to wildfire. Wildfire management agencies must diversify wildfire planning strategies and shift their focus to address these needs and reduce vulnerability. This will require wildfire planners to expand their conception of risk beyond the biophysical and plan for diversified wildfire hazards.

Socially marginalized groups are not a monolith, and planners must consider the distinct needs of each community. Residents who identify with multiple marginalized community groups can face amplified vulnerabilities to disasters due to these overlapping identities [4,5,28,29]. For example, a Latina woman may face heightened vulnerability compared to a white woman, due to the dual hinderances of social marginalization via

ethnicity and gender on her ability to respond to and recover from a wildfire. Wildfire managers and organizations should consider the impacts of these community-specific dynamics. Enhanced understanding of vulnerability, exemplified below, can help managers create more applicable wildfire programming and policy.

### 4.1.1. Age and Vulnerability

Based on the survey results, the 18–24 year-old age group is the least prepared to evacuate during a wildfire, possibly due to the financial and social precarity young adults face as they leave the social safety nets associated with home and school [30]. Since this group was underrepresented in the survey and focus groups, more information is needed to understand their particular vulnerabilities. Partnerships with youth organizations or local schools present opportunities for intervention and deeper engagement.

After young adults, middle-aged respondents (45–54) and older adults (65+) were the least likely to be prepared to evacuate. Perhaps childcare, eldercare, and other family responsibilities are barriers to middle-aged respondents' ability to evacuate. Households with more dependents and young children encounter more difficulties responding to disaster, in part due to the additional strain on household resources [4,10,31]. In the survey, older adults did not report mobility limitations as a barrier to evacuation, which contradicts prevailing research findings on social vulnerability and disasters [10,31]. They indicated that they do not know when to leave or do not have alternative shelter, which supports research showing that populations with a higher proportion of adults over 65 are associated with higher post-disaster shelter needs [31]. Emergency wildfire notifications may not adequately reach these populations, pointing to a need for more targeted and accessible communications.

A deeper analysis of the impact of social capital, or the connections individuals have with others in their community, could be helpful for understanding these vulnerabilities [10]. Social capital may be especially important for socially isolated older adults. Therefore, community programming and engagement efforts that center relationship-building among residents provide a possible solution. Additional interventions include efforts to create strong local social networks that can serve as support systems during a crisis.

### 4.1.2. Gender and Vulnerability

Survey respondents who identify as women are less likely to report being prepared to evacuate during a wildfire. This finding aligns with existing research that indicates that vulnerability disparities along gender lines are due to social inequalities that result in lower wages and the additional care-taking responsibilities that women typically hold [4,10,31]. Women were also significantly more likely to report a multitude of concerns regarding wildfire, possibly due to the additional pressures of caretaking.

Mothers in the Spanish-speaking focus group reported that family size and children with disabilities posed additional challenges to evacuation preparation and evacuating all household members in one vehicle. These findings implicitly reveal that in their community, women bear the responsibility of household disaster planning. In response to this need, wildfire programming should offer additional support to these households. Additionally, wildfire communications materials and educational opportunities should target and accommodate women and caretakers as they are more likely to lead household emergency response.

### 4.1.3. Race, Ethnicity, and Vulnerability

Our findings support research that identifies disproportionate challenges for communities of color during disaster response, as non-white respondents reported less wildfire evacuation preparedness. Racial discrimination and systemic exclusion can impact adaptive capacity and increase vulnerability [6]. Natural disaster response disparities due to race are attributable to racial and ethnic discrimination, inequities in political power and access to social services, and inaccessible disaster communications and recovery fund-

ing [4,10,26,29]. This project's results, coupled with existing literature, imply that wildfire managers and planners must address and compensate for the barriers to resilience caused by social inequity in planning and communications [6].

Survey design issues posed a barrier to a more thorough quantitative analysis of the possible links between ethnicity and wildfire risk. The demographic categories included in the survey match the census to facilitate the comparison to Ventura County demographics. "Hispanic" is an ethnicity category in the U.S. Census, not a race category. The survey did not ask respondents whether they identified as Hispanic or Latine. Thus, this analysis could not include this community's wildfire vulnerabilities from survey data alone. Instead, answers from Spanish-speaking survey respondents and data from the Spanish-speaking focus group results served as proxies to compensate for the missing data. This highlights that focus groups are useful for gathering nuanced, population-specific information about wildfire response.

The Spanish-speaking focus group provided key insights into the barriers to wildfire preparation and response of these groups. These conversations highlighted that a lack of financial resources, coupled with linguistically inaccessible communications, hinder household wildfire preparation and recovery. This finding aligns with literature that demonstrates that the lack of language-appropriate emergency notifications and relief can slow disaster recovery for Latine populations [32,33].

Focus group participants reported social connections with undocumented Mixtec farmworkers in the region and familiarity with the issues they face. These challenges exist at the intersection of citizenship status, language barriers, and ethnic discrimination. The undocumented population in Ventura and Santa Barbara counties is estimated at over 9 percent [5]. Research on wildfire recovery in Ventura shows that Mexican Indigenous (including Mixtec) undocumented communities do not receive adequate communications regarding wildfire threat and do not qualify for federal aid [29,32].

### 4.2. Program and Policy Recommendations

Based on the findings, recommendations for planning and management activities are grouped into the following three categories: (1) an expansion of traditional risk mitigation strategies, (2) approaches to expand community engagement and decision making, and (3) novel approaches that shift the current role of wildfire managers/planners.

#### 4.2.1. Targeted Traditional Strategies

Traditional wildfire risk mitigation strategies at individual and community levels can be adjusted and targeted to better serve communities who are marginalized. Current educational and preparation efforts do not adequately reach all communities who are most vulnerable to wildfire.

Targeted outreach and educational workshops and materials in frequently spoken non-English languages in Ventura, such as Spanish and Mixteco, would make traditional programming more accessible. This communications approach can reach broader audiences and facilitate collaboration among interested and impacted parties for common wildfire planning goals [34]. Furthermore, educational materials such as evacuation checklists are text-heavy and are inaccessible to populations with low literacy rates. Through targeted and linguistically accessible education and engagement strategies for community and household-scale risk mitigation, wildfire organizations can make programming more equitable and build trust with communities facing social inequity/marginalization [32].

Further scoping within these communities is needed to address other potential barriers to participation, such as workshop times and locations. Programs should reduce barriers to participation in communities who are already overburdened by social inequity or are under-resourced by offering food and childcare. Further engagement through existing organizational partnerships could identify specific barriers that then inform more inclusive educational programming.

Finally, most of the vulnerable groups indicated that more effective messaging regarding evacuation orders is needed. Improved evacuation preparation information and communications channels are necessary to help these groups confidently make decisions about their safety. By closing the gaps between current program efforts and these communities' needs, wildfire managers and organizations can reduce vulnerability, and increase evacuation preparedness and response. This would build faster and safer wildfire response among all community members.

4.2.2. Community Engagement Strategies

This research and the work it builds upon are early steps to identifying marginalized communities' needs and barriers to wildfire resilience; expanding programming to meaningfully engage communities outside of top-down planning processes is a necessary next step to ensuring these communities have a voice in wildfire management. Grassroots approaches to community wildfire planning and response efforts can increase residents' capacity to adapt to wildfire and facilitate community–agency collaboration [34,35].

Focus groups and informal community meetings provide opportunities for target audiences to direct the conversation to topics of interest and share nuanced details about their lived experience, as seen in the focus groups conducted in this project. These engagement methods allow managers to ask follow-up questions and build trust with community members. With diversified communication, communities have direct channels for sharing feedback that managers can use for more responsive programming and adaptive management. Our focus group conversations were well received by the community. Direct engagement with specific marginalized groups offers managers and planners the opportunity to identify vulnerabilities and connect residents with the planning process.

Wildfire planning efforts should build community knowledge and capacity by incorporating place-based and community-based participatory research approaches. Participatory research, with appropriate compensation to avoid perpetuating existing inequalities, improves the quality of research findings, builds community skills, and can lead to systemic change in instances of environmental inequity. Community advisory councils are one such strategy for successfully including residents in environmental planning [36]. While community members are aware of wildfire risks and what is needed to reduce their vulnerability, this approach would offer frequent direct contact with the community and provide a bidirectional communications channel for managers and community. By directly participating in information gathering, these communities can ensure diverse needs are addressed. These research strategies require deep engagement with residents and can illuminate how social and environmental factors converge during wildfire events [3]. Wildfire agencies and organizations must directly and intentionally engage historically excluded communities to make wildfire planning truly inclusive.

Management agencies and organizations should thoughtfully consider where along the "consultation" to "empowerment" spectrum community members can feasibly influence planning decisions and transparently communicate that to clarify expectations [37]. Identifying engagement level capacity helps prevent future harm to communities, who could come to distrust managers who express desire to involve them in decision-making but do not have the tools or capacity to integrate feedback or include the community throughout the planning process. Members of the Piru focus group noted that no one had ever come to the community to ask what they needed before, which indicates collaboration is welcome but has been neglected. The most engaging levels of collaboration give community members more agency and decision-making power in the wildfire planning process but require dedicated resources. While the level of engagement managers can support may evolve over time, "empowerment" should be the goal if agencies aim to deeply engage marginalized communities and make system-level change.

Finally, managers should note that meaningful community co-leadership strategies may challenge existing decision-making structures, upset management culture, and shift power away from traditional hierarchies, which are particularly ingrained in the wildfire

community. In some cases, community advisory councils fail to shift decision-making power to impacted communities or lead to systems change [36,38]. It is imperative to prepare leadership and personnel to support more inclusive wildfire planning overall.

4.2.3. Novel Approaches

The feedback provided by research participants implies that novel approaches like community emergency response funds, enhanced collaboration, adopting successful public health models, and intentional efforts to reduce bias are necessary to address the vulnerabilities of marginalized communities. A model for a community emergency relief fund suggested by Spanish-speaking focus group participants is the Central Coast Alliance United for a Sustainable Economy's (CAUSE) 805 UndocuFund, which established a mutual aid fund for undocumented Ventura residents to address income losses related to the Thomas Fire [15]. Similarly, 2023 Senate Bill 227, proposed in the California Senate, would have established this kind of support fund at the state level [39]. These or similar approaches could reduce the impact of wildfire on undocumented communities.

Wildfire planning and response organizations should enhance collaboration with other local agencies, such as transportation and public health, and with non-profit organizations [40]. Collaboration with organizations with similarly aligned goals or target communities helps create community-wide coalitions and allows groups to specialize, rather than requiring wildfire managers to build expertise in all areas. For example, Ventura Fire Safe is developing partnerships with local public health organizations to reduce wildfire risk. Partnerships with social work clinics could support residents by addressing psychological stress and trauma associated with wildfire events, a challenge identified by focus group participants.

Wildfire managers and organizations should adapt successful approaches from other fields, including public health. Promotores (community health workers) programs are utilized in public health and environmental domains to educate residents and change health outcomes. These models rely on community experts who identify with the target population and can serve as peer educators, which is successful because of their cultural proximity to residents [41–43]. This approach could improve wildfire safety education efforts as community trainers have the cultural, linguistic, and community knowledge to effectively reach marginalized communities. Many of the Santa Paula focus group participants are promotores, indicating that there is existing community capacity for this approach in Ventura County and interest in wildfire planning.

There is a need for greater workforce diversity within the emergency response field and for more training. In disaster relief literature on post-hurricane emergency management, research has found that most managers are white and male, which can limit engagement with vulnerable communities [44]. After the 2018 Thomas Fire, emergency services officials were surprised to learn how many local farm workers were Mexican Indigenous. The invisibility of communities can negatively impact post-disaster recovery programs for vulnerable groups, as evidenced in the 1989 Loma Prieta earthquake in California when federal aid workers failed to contract enough bilingual workers [33]. Furthermore, a California state auditor's report indicated that marginalized communities are not fully considered in disaster emergency response [5]. Biases can influence how agencies support community wildfire response [45]; therefore, wildfire managers should consider training on how and why social identity impacts wildfire vulnerability [5]. Public officials and politicians are more responsive to residents with higher socioeconomic status and political pressure can affect response outcomes of wildfire managers, resulting in bias [14,45]. By challenging the biases that persist at the institutional level, managers can expand their ability to effectively engage with the communities that they serve.

*4.3. Considerations for Future Research*

This work is a step towards identifying those most vulnerable to wildfire and mitigating risk; however, future work is needed. The qualitative research methods utilized

here reached a small subset of the Ventura County population and did not fully account for the diverse subsets of the population who may also exhibit unique vulnerabilities to wildfire. For example, the research team did not adequately reach people with disabilities, older adults, and people experiencing homelessness. Furthermore, we did not engage Indigenous Californian communities (primarily Chumash) in the research process. This represents an important gap as these communities have deep ties to the land and wildfire management expertise [46].

## 5. Conclusions

Disparities in wildfire preparedness and response exist based on patterns of social marginalization. This aligns with research indicating that systemic inequities and social marginalization are linked to vulnerability to wildfire, and the disproportionate negative impacts of disasters more broadly. In particular, young adults, women, people of color, residents with disabilities, and newer residents in Ventura County are significantly less likely to be prepared to evacuate. Additionally, people with disabilities and newer residents report facing more barriers to evacuation. Focus group conversations indicated that language barriers exist, preventing Spanish- and Mixtec-speaking communities from receiving the wildfire education and emergency notification communications needed to safely respond. Inequities in wildfire preparation and response highlight opportunities for management agencies and wildfire-focused organizations to address social vulnerability in community wildfire planning and response efforts.

These results and recommendations have planning and policy implications beyond Ventura County and are relevant to broader state and regional wildfire planning, especially as communities respond to increasing wildfire threat and occurrence. This research process and the lessons learned offer a template for agencies and other organizations seeking to gather data on local wildfire vulnerabilities. This work also provides organizations with ideas about how to engage populations who have been marginalized, though each community must evaluate their own unique circumstances, needs, and context. Additionally, the policy and planning recommendations provide examples to other agencies and Fire Safe Councils for adapting traditional planning processes to address social vulnerability. These results and recommendations are adaptable and can be utilized in other communities to disrupt the link between social marginalization and vulnerability to wildfire to equitably increase community resilience across the West.

**Supplementary Materials:** The Supplementary Materials, including survey materials, focus group materials, and additional analyses can be downloaded at: https://www.mdpi.com/article/10.3390/fire7020041/s1.

**Author Contributions:** Conceptualization, B.B., Y.D., I.R.F., A.S., E.O. and S.E.A.; methodology, B.B., Y.D., I.R.F., A.S., E.O. and S.E.A.; software, Y.D. and I.R.F.; validation, B.B., Y.D. and I.R.F.; formal analysis, Y.D. and I.R.F.; investigation, B.B., Y.D., I.R.F., A.S. and E.O.; resources, E.O. and A.S.; data curation, Y.D. and I.R.F.; writing—original draft preparation, I.R.F., E.O. and B.B.; writing—review and editing, B.B., I.R.F., S.E.A. and E.O.; visualization, B.B., Y.D., I.R.F. and A.S.; supervision, B.B., E.O. and S.E.A.; project administration, A.S.; funding acquisition, B.B., Y.D., I.R.F., A.S. and E.O. All authors have read and agreed to the published version of the manuscript.

**Funding:** This research was funded by the Bren School of Environmental Science & Management, WonderLabs, and the DiPaola Foundation. Shefali Juneja Lakhina, PhD, is the co-founder of Wonderlabs, who helped fund this research. While the funder offered guidance and feedback, the funders had no role in the collection of the data, analyses, interpretation, or the writing of the manuscript. This work represents the efforts of the research team, primarily while as students at UCSB, and does not represent the opinions or perspectives of their current employers.

**Institutional Review Board Statement:** The study was determined to be exempt (Category 2) by the Institutional Review Board of the University of California, Santa Barbara, Office of Research (protocol codes 61-21-0442, 61-21-0601, 64-21-0784, 63-21-0714, 61-21-0548, 63-21-0659) because it involved surveys, interviews, and focus groups for which the information obtained was recorded in such a manner that human subjects cannot be identified, either directly or through identifiers linked to the subjects.

**Informed Consent Statement:** Informed consent was obtained from all subjects involved in the study.

**Data Availability Statement:** Due to the reliance on human subjects and in compliance with approved research protocols to protect privacy by the Institutional Review Board of the University of California, Santa Barbara, Office of Research, new data created through this research are not publicly available.

**Acknowledgments:** The authors would like to acknowledge Ventura Regional Fire Safe Council for their partnership in this project and Louis Graup for his support.

**Conflicts of Interest:** The authors declare no conflict of interest.

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
