# Peer review of "Social Inequity and Wildfire Response: Identifying Gaps and Interventions in Ventura County, California"

_fire, doi:10.3390/fire7020041_

Round 1

Reviewer 1 Report

Comments and Suggestions for Authors

I enjoyed very much reading this article. It is very clear methodologically and managed to derive significant results from a non representative sample of the survey, combining with more qualititative focus group approach. It identifies its limitations while pointing towards meaningful directions for research and practice. I just had a minor issue with the Figures 7 and 8, since they refer to male results in captions but they only show results for female and other gender. I think it would be more clear to show the male variable. I think conceptualliy it could be worth to differentiate more the difference between vulnerability and marginalisation and its meanings in wildfire research.

Reviewer 2 Report

Comments and Suggestions for Authors

Thank you for sharing your work. I found  this study to be both interesting and important. Community feedback should be an integral part of developing comprehensive and effective strategies to deal with any natural hazard. Strategies can then be inclusive and protect all members of the community, regardless of socio-economic status or background. An important piece, that could be practically useful ,is the provision of suggestions for improvements that are culturally sensitive and tailored to the specific needs and characteristics of their community.

The manuscript is well written and there is adequate discussion about the perceptions of participants regarding the key aspects considered in your study. 

The one weakness that I find in this manuscript is the limited number of responses that were collected to represent the county and its demographic range. For some questions especially, the number was quite low to allow for any inferences. 

It is my suggestion to try to  expand this study and have a more representative database that will allow you to do a more robust analysis. To you credit, you have identified this issue in the manuscript. 

However, given that this was an initial effort in the area, I still think that the study will be of interest for the journal audience.

I would like to point out one more thing:  the quality of your figures should be improved. Improve the resolution 

Where necessary and for Figure 5 and 7: delete gridlines within figure, improve axis legend resolution for the y axis.
